# Circulating miR-21 as a Potential Biomarker for the Diagnosis of Oral Cancer: A Systematic Review with Meta-Analysis

**DOI:** 10.3390/cancers12040936

**Published:** 2020-04-10

**Authors:** Mario Dioguardi, Giorgia Apollonia Caloro, Luigi Laino, Mario Alovisi, Diego Sovereto, Vito Crincoli, Riccardo Aiuto, Erminia Coccia, Giuseppe Troiano, Lorenzo Lo Muzio

**Affiliations:** 1Department of Clinical and Experimental Medicine, University of Foggia, Via Rovelli 50, 71122 Foggia, Italy; diego_sovereto.546709@unifg.it (D.S.); giuseppe.troiano@unifg.it (G.T.); Lorenzo.lomuzio@unifg.it (L.L.M.); 2Department of Emergency and Organ Transplantation, Nephrology, Dialysis and Transplantation Unit, University of Bari, Via Piazza Giulio Cesare, 70124 Bari, Italy; giorgiacaloro1983@hotmail.it; 3Multidisciplinary Department of Medical-Surgical and Odontostomatological Specialties, University of Campania “Luigi Vanvitelli”, 80121 Naples, Italy; luigi.laino@unicampania.it; 4Department of Surgical Sciences, Dental School, University of Turin, 10127 Turin, Italy; mario.alovisi@unito.it; 5Department of Basic Medical Sciences, Neurosciences and Sensory Organs, Division of Complex Operating Unit of Dentistry, "Aldo Moro" University of Bari, Piazza G. Cesare 11, 70124 Bari, Italy; vito.crincoli@uniba.it; 6Department of Biomedical, Surgical, and Dental Science, University of Milan, 20122 Milan, Italy; riccardo.aiuto@unimi.it; 7Department of Clinical Specialistic and Dental Sciences, Polytechnic University of Marche, Via Tronto 10, 60126 Ancona, Italy; dcermi@virgilio.it

**Keywords:** miR-21, microRNA, oral cancer, OSCC, HNSCC, noncoding RNA

## Abstract

Head and neck squamous cell carcinoma (HNSCC) is one of the main neoformations of the head–neck region and is characterized by the presence of squamous carcinomatous cells of the multi-layered epithelium lining the oral cavity, larynx, and pharynx. The annual incidence of squamous cell carcinoma of the head and neck (HNSCC) comprises approximately 600,000 new cases globally. Currently, the 5-year survival from HNSCC is less than 50%. Surgical, radiotherapy, and chemotherapy treatments strongly compromise patient quality of life. MicroRNAs (miRNAs) are a family of small noncoding endogenous RNAs that function in regulating gene expression by regulating several biological processes, including carcinogenesis. The main upregulated microRNAs associated with oral carcinoma are miR-21, miR-455-5p, miR-155-5p, miR-372, miR-373, miR-29b, miR-1246, miR-196a, and miR-181, while the main downregulated miRNAs are miR-204, miR-101, miR-32, miR-20a, miR-16, miR-17, and miR-125b. miR-21 represents one of the first oncomirs studied. The present systematic review work was performed based on the preferred reporting items for systematic review and meta-analysis (PRISMA) protocol. A search was carried out in the PubMed and Scopus databases with the use of keywords. This search produced 628 records which, after the elimination of duplicates and the application of the inclusion and exclusion criteria, led to 7 included articles. The heterogeneity of the studies according to the odds ratio was high, with a Q value of 26.616 (*p* < 0.001), and the *I^2^* was 77.457% for specificity. The heterogeneity was high, with a Q value of 25.243 (*p* < 0.001) and the *I^2^* was 76.231% for sensitivity. The heterogeneity of data showed a Q value of 27.815 (*p* < 0.001) and the *I^2^* was 78.429%. Therefore, the random-effects model was selected. The diagnostic odds ratio was 7.620 (95% CI 3.613–16.070). The results showed that the sensitivity was 0.771 (95% CI 0.680–0.842) (*p* < 0.001) while, for specificity, we found 0.663 (95% CI 0.538–0.770) (*p* < 0.001). The negative likelihood ratio (NLR) was 0.321 (95% CI 0.186–0.554), and the positive likelihood ratio (PLR) was 2.144 (95% CI 1.563–2.943). The summary ROC plot demonstrates that the diagnostic test presents good specificity and sensitivity, and the area under the curve (AUC), as calculated from the graph, was 0.79.

## 1. Introduction

Head and neck squamous cell carcinoma (HNSCC) is the most common histologic neoplasm arising in the head–neck region, and is characterized by the presence of squamous carcinomatous cells in the multi-layered epithelium lining the oral cavity, larynx, and pharynx. The annual incidence of HNSCC comprises approximately 600,000 new cases all over the world [1]. HNSCCs are divided according to their main groups of oral squamous cancer cell carcinoma (OSCC), larynx squamous cell carcinoma (LSCC), and oropharynx squamous cell carcinoma (OPSCC). Among these, OSCC represents the neoplasm with the highest incidence in the head and neck region. In Italy, the average incidence of OSCC is 8.44 new cases per every 100,000 inhabitants, with a higher frequency among men than women. Mortality rates in Europe peaked in the 1980s, with 6.3 cases per 100,000 people [2].

Currently, the 5-year survival of patients with HNSCC is less than 50%, and the most commonly applied therapies, including surgery, radiotherapy, and chemotherapy, strongly compromise patient quality of life. Tobacco smoke and alcohol are the main risk factors for squamous cell carcinoma; researchers have estimated that around 80% of carcinomas are due to these factors [3,4]. Scientific evidence suggests the importance of other factors among the mechanisms of carcinogenesis, including infectious agents (papillomavirus) traumatic agents, nutritional factors, genetic factors, ultraviolet radiation, and immunosuppression [5].

There are several genetic alterations involved in oral carcinogenesis. Among these, alterations in oncosuppressors (APC, p53), proto-oncogenes (Myc), oncogenes (Ras), and genes that control normal cellular processes (EIF3E, GSTM1) play a fundamental role in cancer development [6]. Events such as DNA methylation, histone modifications, and noncoding alterations of RNA (e.g., microRNAs (miRNAs)) are also involved in the onset and progression of oral cancer [7].

MicroRNAs (miRNAs) are a family of small noncoding endogenous RNAs that function in influencing gene expression by regulating a number of biological processes, including carcinogenesis [8]. They are transcribed by RNA polymerase II and III, and this generates a series of precursors of different lengths, which undergo various enzymatic cleavage processes to finally form the mature miRNA [9].

Mature miRNA is incorporated into an RNA-induced silencing protein complex called RISC. RISC loaded with miRNA is able to mediate gene silencing through transcript degradation or inhibition mechanisms, based on the degree and nature of the complementarity between the miRNA and target messenger RNA (mRNA) [10].

The biogenesis of miRNA is regulated at various levels: miRNA transcription, transport, and RISC binding and targeted degradation. In recent decades, researchers have found that miRNA expression is dysregulated in human malignant tumor cells [11].

The first evidence in this regard was provided by the group of Croce from studies on chronic lymphatic leukemia B cells, identifying the suppression or downregulation of two miRNA genes (MIR-15a and MIR-16-1), which act as tumor suppressors by inducing the apoptosis of malignant tumor cells [12].

Dysregulation mechanisms include chromosomal abnormalities such as those resulting from amplification or gene deletion, alterations of transcriptional control, epigenetic alterations, and defects of proteins involved in the biogenesis of miRNAs [13].

The dysregulation of a miRNA allows the cancer cell to develop specific cancer capacities including support of the proliferative signal with avoidance of the suppressing mechanisms, resistance to apoptosis, invasion and metastasis capabilities, and the induction of angiogenesis [14].

A previous meta-analysis from our group reported that dysregulated miRNA expression in tissue can be used as a predictor of worse prognosis in OSCC patients. In particular, upregulated miRNAs with prognostic capabilities were miR-21, miR-455-5p, miR-155-5p, miR-372, miR-373, miR-29b, miR-1246, miR-196a, and miR-181, while the downregulated miRNAs were miR-204, miR-101, miR-32, miR-20a, miR-16, miR-17, and miR-125b [15]. The gene sequence of MIR-21 is 5 ′ TAGCTTATCAGACTGATGTTGA 3 ′, and it is located at chromosome 17q23.1 [16,17].

More recently, a study on the plasma of OSCC patients confirmed the diagnostic and prognostic potential of miR-21 expression [18]. Subsequently, Karimi et al. reported that serum levels of miR-21, miR–24, and miR-29a can be used as markers for the detection of carcinoma and, therefore, also potentially used to develop new therapeutic strategies [19].

Hence, knowledge of the expression and presence of miR-21 in serum and plasma has potential diagnostic value and can be of help to clinicians, especially if its presence can be used as a predictive and prognosis biomarker for HNSCC. The aim of this systematic review and meta-analysis was to investigate and summarize results in the literature, regarding the potential use of circulating miR-21 in plasma or serum as a diagnostic biomarker for OSCC patients.

## 2. Materials and Methods

The following systematic review was conducted according to guidelines reported in the indications of the preferred reporting items for systematic review and meta-analysis (PRISMA) [20].

This study aimed to answer the following PICO question: “What are the expression levels of Mir-21 in the blood of patients with head and neck squamous cell carcinoma (HNSCC) compared to patients without oral cancer? In addition, can miR-21 circulation serve as a potential biomarker for early oral cancer diagnosis?” The following scheme was applied: population (patients with oral cancer), intervention (circulating miR-21), control (patients who do not suffer from cancer), and outcome (miR-21 expression detectable in blood of patients with oral cancer).

### 2.1. Eligibility Criteria

After initial phase screening abstracts were identified in online databases, the potentially eligible articles were qualitatively evaluated, in order to investigate the role of miR-21 as a biomarker for oral cancer.

The studies taken into consideration were only clinical studies (systemic reviews were taken into consideration as a bibliographic source to reveal studies potentially acceptable for inclusion), which dealt with the topic of microRNAs in relation to oral cancer. Specifically, we sought studies evaluating the expression of miR-21 in patients with HNSCC, compared to healthy controls published in recent years in the English language. We focused on articles published in the last 20 years, as studies on miRNAs started in early 2000.

The inclusion and exclusion criteria we applied for the full text analysis were the following:Studies investigating the expression in whole blood, plasma, or serum of miR-21 in OSCC patients with oral cancer and healthy controls.Articles reporting data related to the diagnostic prediction and predictive performance, including parameters such as specificity, sensitivity, and area under the receiver operating characteristic (ROC) curve.Case reports, reviews, and in vitro studies, as well as studies on animal models and human cell lines were excluded from this study.We excluded all articles that did not focus on the expression of miR-21 or that did not report adequate data related to this.

### 2.2. Research Methodology

Studies were identified through the bibliographic searching of electronic databases. The literature search was conducted using the search engines PubMed and Scopus. The searches were conducted between 1/01/2020 and 09/02/2020, and the last search for a partial update of the literature was conducted on 10/02/2020. The details regarding the search terms and combination strategies used in the literature research are reported in Table 1.

### 2.3. Screening Methodology

The obtained search records were subsequently examined by two independent reviewers (M.D. and D.S.), and a third reviewer (G.T.) acted as a decision maker in cases of disagreement between the two reviewers. The screening included the analysis of the title and the abstract to eliminate records not related to the topics of the review. After the screening phase, duplicates were removed and the complete texts of the articles were analyzed, from which we chose the ones eligible for qualitative analysis and inclusion in the meta-analysis. The data sought by the two reviewers in the included studies are as follows.

Primary outcome: the reported data on the sensitivity, specificity, and/or area under the curve (AUC) for the detection of Mir-21 in circulation in patients with HNSCC and healthy controls.

The K agreement between the two screening reviewers was 0.8584 (Table 2) [21]. The K agreement was based on the formulas of the Cochrane Handbook for Systematic Reviews [22].

The entire selection and screening procedures are described in the flow chart (Figure 1).

### 2.4. Statistical Analysis Protocol

The protocol with which the meta-analysis was performed was based on the indications of the Cochrane Handbook for Systematic Reviews of Diagnostic Test Accuracy. The accuracy of the test is important. As reported in Section 10.1.4.1 of the Cochrane Handbook for Systematic Reviews of Diagnostic Test Accuracy, the meta-analysis performed was limited to characterizing the accuracy of a single test (mirR-21 in the blood in the diagnosis of HNCSS in this specific case) and aimed to estimate an average synthesis value of sensitivity and specificity, as well as to describe how the sensitivity and specificity vary with the variation of the threshold, by estimating an ROC curve of summary.

During data extraction of the included studies, the cut-off threshold that indicates positivity or negativity in the diagnostic test was identified for each study, evaluating the possibility of comparability between the different studies (Cochrane Handbook for Systematic Reviews of Diagnostic Test Accuracy Section 10.4.1). The extracted data, performed by several independent auditors, has been reported and summarized in tables, and the data were not subject to statistical formulae, but reported in tables in the results section, and subsequently inserted in programs for statistical analysis (Section 10.3.5).

The meta-analysis models were used to calculate aggregate sensitivity, specificity, positive likelihood ratio (PLR), negative likelihood ratio (NLR), and diagnostic odds ratio (DOR). As software for meta-analysis, we decided to use Reviewer Manager 5.3 (Cochrane collaboration, Copenhagen, Denmark) [23], for graphic representation (summary ROC) and to help in the drafting of the meta-analysis work, and Open Meta-Analyst version 10 (Tufts University, Medford, MA, USA) for all statistical analyses, as Reviewer Manager is not suitable for the calculation of all analyses, as reported in Chapter 10. Paragraph 5.2 of the Cochrane Handbook for Systematic Reviews of Diagnostic Test Accuracy.

The representation of the final data resulting from the meta-analysis was reported both graphically (forest plot and summary ROC) and by reporting the data. Summary ROC (SROC) curves, which summarize the sensitivity and specificity of each study with regard to evaluating the diagnostic effects, were also calculated, while the AUC (area under the curve) was obtained starting from the SROC chart using ImageJ software [24]. In addition, the heterogeneity was evaluated through the Q test and *I**^2^*. A P value lower than 0.05 for the Q test or *I^2^* larger than 0.50 were considered as the thresholds for the presence of significant heterogeneity. The risk of bias in the studies was calculated following the guidelines reported in the Quality Assessment of Diagnostic Accuracy Studies 2 (QUADAS-2) [25].

## 3. Results

In total, 628 manuscripts were identified on PubMed and Scopus. After eliminating the duplicate articles and applying the inclusion and inclusion criteria, the following articles were obtained for further investigation: Hsu et al. 2012 [26], Liu et al. 2013 [27], Ren et al. 2014 [28], Mahmood et al. 2019 [18], Karimi et al. 2020 [19], Lu et al. 2019 [29], and Ishinaga et al. 2019 [30].

### 3.1. Study Characteristics and Data Extraction

The data extracted from the included studies are summarized in Table 3.

### 3.2. Risk of Bias

Results of the risk of bias evaluation assessed using the QUADAS-2 scale are shown in detail in Table 4. For each analyzed category, a judgement of: low risk, high risk, or unclear risk was reported after evaluation by two authors in a joint session. The risk of bias among the studies was calculated by assessing the heterogeneity through the *I^2^* and Q value. The heterogeneity was high with an *I^2^* value of 80.73% for the DOR, representing a limit for this revision. A meta-analysis of the diagnostic odds ratio showed the presence of high heterogeneity with a Q value of 26.616 (*p* < 0.001) and *I^2^* of 77.457%. Similar results were obtained for specificity; in fact the heterogeneity was high with a Q value 25.243 (*p* < 0.001) and an *I^2^* of 76.231%. Sensitivity was shown by a Q value of 27.815 (*p* < 0.001) and an *I^2^* of 78.429%. Therefore, a random-effects model was applied to each meta-analysis.

### 3.3. Meta-Analysis: Diagnostic Accuracy of miR-21 for OSCC and HNSCC

The aggregate diagnostic odds ratio (DOR) was 7.620 (95% CI 3.613–16.070) (Figure 2). Results show an aggregate sensitivity of 0.771 (95% CI 0.680–0.842) (*p* < 0.001), while for specificity, we found an aggregate value of 0.663 (95% CI 0.538 0.770) (*p* < 0.001), as shown in Figure 3. The NLR was 0.321 (95% CI, 0.186–0.554) and the PLR was 2.144 (95% CI, 1.563–2.943) (Figure 4). The summary ROC plot shows how the diagnostic aggregate value of the AUC was 0.79 (Figure 5).

## 4. Discussion

Currently, more than 120 publications have investigated miR-21 expression in HNSCC patients in the literature. miR-21 appears to be one of the main miRNAs associated with OSCC. The upregulation of miR-21 was demonstrated for several cancers, including glioblastomas and breast, colon, lung, pancreatic, thyroid, and ovarian cancer.

For the diagnosis of HNSCC, clinicians applied both clinical examination and diagnostic tools, such as CT scans followed by a biopsy of the lesion. However, such methods are more invasive and, in this context, the addition of a noninvasive diagnostic test based on the analysis of a marker on plasma or serum would be beneficial in the early diagnosis of such diseases. Many studies show that circulating miRNAs can act as biomarkers for the detection of cancer in early stages, as circulating miRNAs are expressed abnormally in HNSCC patients.

Being among the first to investigate the presence of miR-21 in plasma in 2012, Hsu et al. analyzed 50 HNSCC patients and 30 healthy controls, reporting a sensitivity of 83.3% and a specificity of 51.9% with an AUC of 0.741, suggesting that the presence of miR-21 could be used as a biomarker in the plasma of HNSCC patients [27]. In 2014, Ren et al. reported in a study on circulating miR-21 and PTEN that the sensitivity and specificity of the diagnostic test for miR-21 expression were 62.1% and 90.6%, respectively, with an AUC equal to 0.788 [28]. Similarly, in a prognostic and predictive study of 100 oral cancer patients with 100 controls in 2019, Mahmood et al. found a sensitivity of 91% and a specificity of 54% for circulating miR-21, with an AUC value of 0.829 [18].

In a longitudinal study on circulating miR-21 carried out on 86 HNSCC patients and 29 healthy controls conducted by Ishinaga et al. in 2019, the value for the area under the curve (AUC) for plasma miR-21 was 0.756. In addition, data for the sensitivity, specificity, and positive and negative predictive values were 70.9%, 69.0%, 87.1%, and 44.4% respectively.

In 2020, Karimi et al. published the most recent study included in this review, investigating the expression of miR-21, miR-24, and miR-29a in the serum of patients with oral squamous cell carcinoma (20 patients with OSCC and 20 controls), reporting values equal to 95% for both the sensitivity and specificity with respect to miR-21. The weakness of this study was the low sample size. In fact, these data were not in agreement with the previous studies, as is also shown in the summary ROC (Figure 5) [19].

In 2019, Lu et al. published a study including a cohort of 82 oral cancer patients and 53 healthy subjects and their results differed from previous studies and showed opposing trends to the data presented by Karimi et al. 2020 (as represented in the summary ROC in Figure 5), with an AUC of 0.579 and a sensitivity and specificity of 64.2% and 46.3%, respectively [29].

The meta-analysis of the included studies gives us interesting diagnostic accuracy results with an area under the curve of 0.79 and values of sensitivity and specificity at 0.771 and 0.663, respectively. These data reported a worse diagnostic accuracy than a previous systematic review conducted by Wu et al. in 2014, which reported accuracy data with a sensitivity level of 0.78, specificity of 0.82, PLR of 4.4, NLR of 0.26, DOR of 17, and AUC of 0.87. Wu et al. focused on multiple types of cancer, not including HSNCC and OSCC, for the small number of studies conducted on miR-21 on those particular types of neoformations. To date, this meta-analysis is the first that examines the diagnostic accuracy of circulating miR-21 on oral cancer and head and neck neoplasms [31].

### Limits of the Study

The limitations of this study include the low number of included studies, the inconsistent identification of an equal cut-off level for circulating miR-21 in all the studies, and the diversity between the serum and plasma used in the included studies, as well as the high heterogeneity among the selected studies. These biases may partially limit the results obtained from the meta-analysis.

## 5. Conclusions

In conclusion, our data provide evidence, notwithstanding the limitations of our review, that circulating miR-21 can be used as a diagnostic biomarker in the future, and we hope that better knowledge of the serum or plasma expression of miR-21 will allow it to be used as a prognostic and predictive biomarker for HNSCC patients.

## Figures and Tables

**Figure 1 cancers-12-00936-f001:**
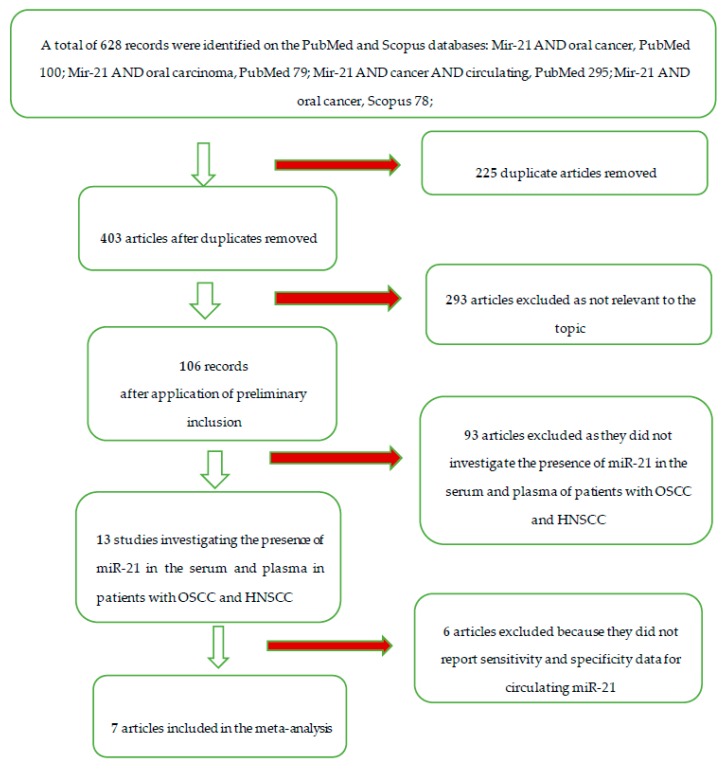
Flow chart of the different phases of the systematic review. From a total of 628 records, we ended up with 13 clinical studies investigating miR-21 in patients with HNSCC, and 6 articles that assessed sensitivity and specificity by satisfying the inclusion criteria.

**Figure 2 cancers-12-00936-f002:**
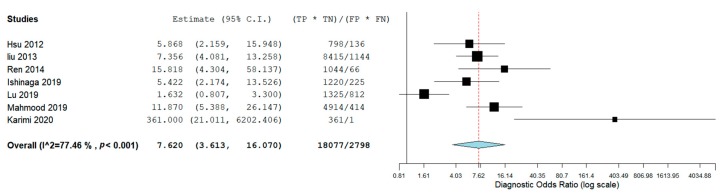
Diagnostic odds ratio.

**Figure 3 cancers-12-00936-f003:**
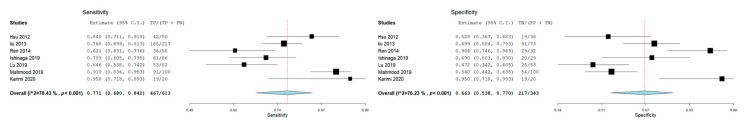
Forest plot of the sensitivity and specificity.

**Figure 4 cancers-12-00936-f004:**
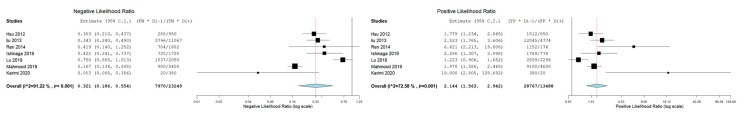
Forest plot of the positive likelihood ratio and negative likelihood ratio.

**Figure 5 cancers-12-00936-f005:**
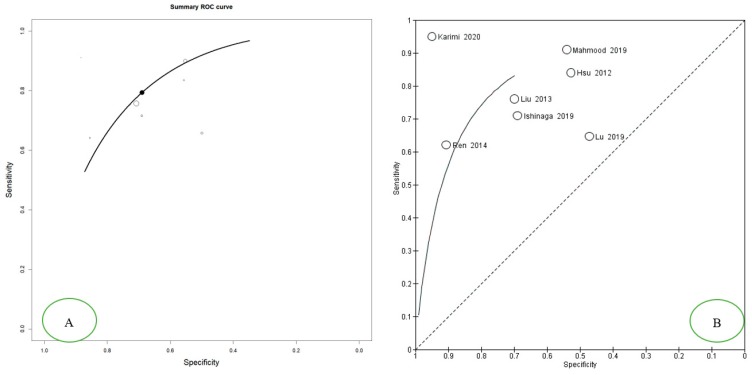
The Summary ROC curve is intended to summarize the relationship between the TPR (true positive rate—sensitivity) and FPR (false positive rate—specificity) across the set of studies. (**A)** Summary ROC Curve Open Meta-Analyst version 10, the optimal position between the TPR and FPR response to individual studies is highlighted on the SROC with a black point; (**B)** Summary ROC Curve Reviewer Manager 5.3, the position of the individual studies with respect to the Summary ROC is highlighted.

**Table 1 cancers-12-00936-t001:** Complete overview of the search methodology. Records identified by databases: 628.

Database Provider	Keywords	Search Details	Number of Records	Articles after Removal of Duplicates	Number of Records) after Restriction by Year of Publication (Last 20 Years)	Number of Articles that have Investigated the Expression of mir-21 in Relation to Oral Cancer	Number of Articles Investigating the Presence in Serum and Plasma of mir-21, in Patients with Oral Squamous Cancer Cell Carcinoma (OSCC) and Head and Neck Squamous Cell Carcinoma (HNSCC)	Number of Articles included in the Meta-Analysis
**PubMed**	Mir-21 AND oral cancer	Mir-21[All Fields] AND ("mouth neoplasms"[MeSH Terms] OR ("mouth"[All Fields] AND "neoplasms"[All Fields]) OR "mouth neoplasms"[All Fields] OR ("oral"[All Fields] AND "cancer"[All Fields]) OR "oral cancer"[All Fields])	100					
**PubMed**	Mir-21 AND oral carcinoma	Mir-21[All Fields] AND (("mouth"[MeSH Terms] OR "mouth"[All Fields] OR "oral"[All Fields]) AND ("carcinoma"[MeSH Terms] OR "carcinoma"[All Fields]))	79					
**PubMed**	Mir-21 AND cancer AND circulating	Mir-21[All Fields] AND ("neoplasms"[MeSH Terms] OR "neoplasms"[All Fields] OR "cancer"[All Fields]) AND circulating [All Fields]	299					
**Scopus**	Mir-21 AND oral cancer	TITLE-ABS-KEY (mir-21 AND oral AND cancer)	78					
**Scopus**	Mir-21 AND oral carcinoma	TITLE-ABS-KEY (mir-21 AND oral AND carcinoma)	72					
**Total**			628	403	403	106	13	7

**Table 2 cancers-12-00936-t002:** K agreement calculation, Po = 0.9339 (proportion of agreement), Pe = 0.809 (agreement expected). K agreement = 0.654 (no agreement), 0.0–0.20 (slight agreement), 0.21–0.40 (fair agreement), 0.41–0.60 (moderate agreement), 0.61–0.80 (substantial agreement), and 0.81–1.00 (almost perfect agreement). The K agreement was calculated from the 105 articles to include five articles with the application of the inclusion and exclusion criteria.

/	/	Reviewer 2	Reviewer 2	Reviewer 2	/
		Include	Exclude	Unsure	Total
**Reviewer 1**	Include	7	0	0	7
**Reviewer 1**	Exclude	0	92	2	94
**Reviewer 1**	Unsure	1	4	0	5
	Total	8	96	2	106

**Table 3 cancers-12-00936-t003:** The data extracted for the six articles included in the meta-analysis.

Autor, Date, Journal	MicroRNA Investigated	Type of Sample	Gender of Patients	Age	Risk Factors	Country	Carcinoma	Control Group	Sensitivity miR-21	Specificity miR-21	AUC	Cut-off Value
Karimi et al. 2020, *J Oral Pathol Med* [29]	miR-21, miR-24, miR-29a	Serum	Male (14 OSCC-14 Control) Female (6 OSCC-6 Control)	OSCC (46.60 ± 10.69); Control 47.10 ± 17.66	Smoking condition (OSCC 10- Control 10)	Iran	OSCC (*n* = 20)	(*n* = 20)	95% (CI: 76.39–99.11)	95 % (CI: 76.39–99.11)	\	\
Ishinaga et al. 2019, *Carcinogenesis* [31]	miR-21	Plasma	Male (77 HNSCC-23 Control) Female (9 HNSCC-6 Control)	HNSCC (65.5 ± 11.2) Control (61.3 ± 10.6)	/	Japan	HNSCC (*n* = 86)	(*n* = 29)	70.9%	69.0%	0.756 (95% CI: 0.661–0.851	1.15
Mahmood et al. 2019, Pak *J Med Sci.* [18]	miR-21	Plasma	136 Males and 64 Females	32.29 ± 4.98 for males, 31.77± 5.4 for female.	Smoking status (66 Yes, 34 No)	Pakistan	Oral cancer (*n* = 100)	(*n* = 100)	91%	54%	0.829	35 Ct
Lu et al. 2019, *Molecular Therapy—Nucleic Acids* [30]	miR-99a-5p, miR31-5p, miR-138-5p, miR-21-5p, miR-375-3p	Serum	Male (OCSS 61, Control 27) Female (OSCC 21, Control 26)	Age ≥ 60 41 (50%) 12 (22.6%) Age < 60 41 (50%) 41 (77.4%)	/	China	Oral cancer (*n* = 82)	(*n* = 53)	64.2%	46.3%	0.579 (95% CI: 0.483-0.675)	\
Ren et al. 2014, *Biomarkers* [28]	miR-21	Blood	Male 39 (67.2%); Female 19 (32.8%)	Age Mean (range) 61 (25–92)	Nonsmokers 31 (53.4%). Current smokers 27 (46.6%)	China	OSCC (*n* = 58)	(*n* = 32)	62.1%	90.6%	0.788 (95% CI: 0.692–0.883)	9.646
Liu et al. 2013 *Cancer Biol Ther* [26]	miR-16, miR -21, miR -24, miR -155	Plasma	NPC 149 males and 68 females; Control group 50 males and 23 females	NPC, age 45.93 ± 11.78; Control, Age 40.08 ± 10.79.	/	China	Nasopharyngeal carcinoma (NPC) (*n*= 217)	(*N* = 73)	76%	69,9	79.2% DS 0.030(95% CI: 0.733 0.852)	\
Hsu et al. 2012, *Tumor Biol* [27]	let-7a, miR-21, miR26b, miR-34c, miR-99a, miR133a, miR-137, miR-184, miR-194a, miR-375	Plasma	HNSCC (48 Male, 2 Female); Control (27 Male, 9 Female)	HNSCC: aged 34–82 years (mean ± SD, 54.61 ± 10.38); Control: aged 23–62 years (mean ± SD, 36.23 ± 2.34)	/	Taiwan	HNSCC (*n* = 50)	(*n* = 36)	83.3%	51.9%	0.741	\

**Table 4 cancers-12-00936-t004:** Quality Assessment of Diagnostic Accuracy Studies 2 (QUADAS-2).

Study	Risk of Bias				Applicability Concerns		
	Patient Selection	Index Test	Reference Standard	Flow and Timing	Patient Selection	Index Test	Reference Standard
**Karimi et al. 2020, *J Oral Pathol Med*** [29]	**L**	**L**	**U**	**L**	**L**	**U**	**L**
**Ishinaga et al. 2019, *Carcinogenesis*** [31]	**L**	**U**	**L**	**L**	**L**	**L**	**L**
**Mahmood et al. 2019, Pak *J Med Sci.*** [18]	**L**	**L**	**L**	**L**	**L**	**L**	**L**
**Lu et al. 2019, *Molecular Therapy—Nucleic Acids*** [30]	**L**	**U**	**L**	**L**	**L**	**L**	**L**
**Ren et al. 2014, *Biomarkers*** [28]	**U**	**L**	**L**	**L**	**L**	**L**	**U**
**Liu et al. 2013 *Cancer Biol Ther*** [26]	**L**	**L**	**L**	**L**	**L**	**L**	**L**
**Hsu et al. 2012, *Tumor Biol*** [27]	**H**	**L**	**L**	**L**	**L**	**L**	**L**

L = Low risk, H = high risk, U = unclear risk.

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
