# Peer review of "Circulating miR-21 as a Potential Biomarker for the Diagnosis of Oral Cancer: A Systematic Review with Meta-Analysis"

_cancers, 2020, doi:10.3390/cancers12040936_

Round 1

Reviewer 1 Report

This article sum up the studies that have evaluated the role of MiR21 in the oral cancer onset. The revision was interesting and quite update. Here are a few tips that can enrich your work. Usually a signature of microRNAs is used, not just one, to obtain a panel of microRNAs to tested and validated, in the clinical utility are recommended more microRNAs individually or in combination will be tested. However, the exclusion criteria is not well delineated. The difference between serum and plasma used in the included studies are not specify. Additionally, in table 3 should be added gender of patients recruited in the studies, risk factors and age of study population. Add the origin from which countries around the world come the patients come. Is not clear that also considered studies that only reported results from in vitro, in silico or in vivo studies. It was considered a quality assessment tools?

Author Response

Reviewer 1

This article sum up the studies that have evaluated the role of MiR21 in the oral cancer onset. The revision was interesting and quite update. Here are a few tips that can enrich your work. Usually a signature of microRNAs is used, not just one, to obtain a panel of microRNAs to tested and validated, in the clinical utility are recommended more microRNAs individually or in combination will be tested. However, the exclusion criteria is not well delineated. The difference between serum and plasma used in the included studies are not specify. Additionally, in table 3 should be added gender of patients recruited in the studies, risk factors and age of study population. Add the origin from which countries around the world come the patients come. Is not clear that also considered studies that only reported results from in vitro, in silico or in vivo studies. It was considered a quality assessment tools?

Answer

thanks for reviewing the manuscript and the suggestions and advice given

  • I specified whether serum or plasma was used in the included studies
  • Table n3 has been improved by adding age, gender, risk factor and country
  • Only in vivo clinical trials were included
  • Quadas 2 was used to evaluate the studies.

Reviewer 2 Report

In this review article 'miR-21 circulating as a potential biomarker for the diagnosis of oral cancer: systematic review with meta-analysis' authors propose that circulating miR-21 can be used as a potential diagnostic biomarker for HNSCC. Though this meta-analysis is the first report, authors have based their conclusions on very few studies.

There are few queries that need to be addressed-

  1. Why only 7 articles were considered out of 13, why not all?
  2. The entire manuscript should be thoroughly edited for grammar, sentence formation.
  3. The title should have been- 'Circulating miR-21 as potential ---'
  4. In the discussion section, HNCSS should be HNSCC.
  5. The statistics performed as per different guidelines and models should be discussed elaborately.
  6. The reviewer is unable to understand what to infer from Figure 5 A and B, results should be discussed with the reasoning behind doing that particular analysis, rather than mentioning the values 

Author Response

Reviewer 2

In this review article 'miR-21 circulating as a potential biomarker for the diagnosis of oral cancer: systematic review with meta-analysis' authors propose that circulating miR-21 can be used as a potential diagnostic biomarker for HNSCC. Though this meta-analysis is the first report, authors have based their conclusions on very few studies.

There are few queries that need to be addressed-

  1. Why only 7 articles were considered out of 13, why not all?
  2. The entire manuscript should be thoroughly edited for grammar, sentence formation.
  3. The title should have been- 'Circulating miR-21 as potential ---'
  4. In the discussion section, HNCSS should be HNSCC.
  5. The statistics performed as per different guidelines and models should be discussed elaborately.
  6. The reviewer is unable to understand what to infer from Figure 5 A and B, results should be discussed with the reasoning behind doing that particular analysis, rather than mentioning the values 

Answer

thanks for reviewing the manuscript and the suggestions and advice given.

  1. For greater clarity or redesigned the flow chart, improving the description of the criteria with which the articles were excluded. It went from 13 articles to 7, because six articles did not report sensitivity and specificity data (figures 1)
  2. I had the article reviewed by MDPI's English editing service
  3. Changed the title as recommended
  4. Changed from HNCSS to HNSCC
  5. New chapter inserted -4. Statistical analysis protocol-
  6. Added explanation figure 5

Round 2
